## [Peer Review File · EMBO Reports]

Hyperactivation of mTORC1 blocks stem cell fate transitions through TFE3-NuRD association

Xueting Xu, Peizhi Li, Shuhui Xu, Lulu Wang, Xinyu Wu, Yin Gao, Tanveer Ahmed, Yinghua Huang, Dajiang Qin, and Baoming Qin

Corresponding author(s): Xueting Xu (xu_xueting@gibh.ac.cn) , Lulu Wang (wang_lulu@gzhmu.edu.cn), Baoming Qin (qinbaoming@genomics.cn)

Review Timeline:

Submission Date:	10th Nov 24
Editorial Decision:	4th Dec 24
Revision Received:	4th May 25
Editorial Decision:	6th Jun 25
Revision Received:	5th Jul 25
Accepted:	25th Jul 25

Editor: Achim Breiling

Transaction Report:

Dear Dr. Xu,

Thank you for the transfer of your manuscript to EMBO reports. I have now received the reports from the three referees that were asked to evaluate your study, which can be found at the end of this email.

As you will see, the referees think that these findings are of interest. However, they have several comments, concerns, and suggestions, indicating that a major revision of the manuscript is necessary to allow publication of the study in EMBO reports. As the reports are below, and all the referee concerns need to be addressed, I will not detail them here.

Given the constructive referee comments, I would like to invite you to revise your manuscript with the understanding that the concerns of the referees must be addressed in the revised manuscript and in a detailed point-by-point response. Acceptance of your manuscript will depend on a positive outcome of a second round of review. It is EMBO reports policy to allow a single round of revision only and acceptance of the manuscript will therefore depend on the completeness of your responses included in the next, final version of the manuscript.

- 1) a .docx formatted version of the final manuscript text (including legends for main figures, EV figures and tables), but without the figures included. Figure legends should be compiled at the end of the manuscript text.
- 2) individual production quality figure files as .eps, .tif, .jpg (one file per figure), of main figures and EV figures. Please upload these as separate, individual files upon re-submission.

- 4) a complete author checklist, which you can download from our author guidelines (<https://www.embopress.org/page/journal/14693178/authorguide>). Please insert page numbers in the checklist to indicate where the requested information can be found in the manuscript. The completed author checklist will also be part of the RPF.

- 5) that primary datasets produced in this study (e.g. RNA-seq, ChIP-seq, structural and array data) are deposited in an

appropriate public database. If no primary datasets have been deposited, please also state this in a dedicated section (e.g. 'No primary datasets have been generated and deposited'), see below.

The accession numbers and database should be listed in a formal "Data Availability" section that follows the model below. This is now mandatory (like the COI statement). Please note that the Data Availability Section is restricted to new primary data that are part of this study. This section is mandatory. As indicated above, if no primary datasets have been deposited, please state this in this section

Data availability

8) Regarding data quantification and statistics, please make sure that the number "n" for how many independent experiments were performed, their nature (biological versus technical replicates), the bars and error bars (e.g. SEM, SD) and the test used to calculate p-values is indicated in the respective figure legends (also for EV and Appendix figures). Please also check that all the p-values are explained in the legend, and that these fit to those shown in the figure. Please provide statistical testing where applicable. Please avoid the phrase 'independent experiment', but clearly state if these were biological or technical replicates. Please also indicate (e.g. with n.s.) if testing was performed, but the differences are not significant. In case n=2, please show the data as separate datapoints without error bars and statistics. See also: <http://www.embopress.org/page/journal/14693178/authorguide#statisticalanalysis>

9) Please add scale bars of similar style and thickness to microscopic images, using clearly visible black or white bars (depending on the background). Please place these in the lower right corner of the images themselves. Please do not write on or near the bars in the image but define the size in the respective figure legend.

10) Please also note our reference format:

12) We now use CRedit to specify the contributions of each author in the journal submission system. CRedit replaces the author contribution section. Please use the free text box to provide more detailed descriptions and do NOT provide your final manuscript text file with an author contributions section. See also our guide to authors: <https://www.embopress.org/page/journal/14693178/authorguide#authorshipguidelines>

13) All Materials and Methods need to be described in the main text using our 'Structured Methods' format, which is required for

all research articles. According to this format, the Methods section should include a Reagents and Tools Table (listing key reagents, experimental models, software, and relevant equipment and including their sources and relevant identifiers), uploaded as separate file, and a Methods section in which we encourage the authors to describe their methods using a step-by-step protocol format with bullet points, to facilitate the adoption of the methodologies across labs. More information on how to adhere to this format as well as downloadable templates (.doc) for the Reagents and Tools Table can be found in our author guidelines (section 'Structured Methods'):

14) Please order the manuscript sections like this, using these names:

Title page - Abstract - Keywords - Introduction - Results - Discussion - Methods - Data availability section - Acknowledgements (including funding information) - Disclosure and Competing Interests Statement - References - Figure legends - Expanded View Figure legends

15) Please make sure that all the funding information is also entered into the online submission system and that it is complete and similar to the one in the acknowledgement section of the manuscript text file.

I look forward to seeing a revised form of your manuscript when it is ready.

Yours sincerely,

Referee #1:

Li et al demonstrate that hyperactivation of mTOR signalling and nuclear localisation of TFE3 promote self-renewal and inhibit the exit from pluripotency, consistent with prior findings. Additionally, hyperactivation of mTOR signalling and nuclear localisation of TFE3 prevent somatic cell reprogramming. To explore TFE3's nuclear functions, the authors analysed published IP-MS data for TFE3 interactors and confirmed its direct interaction with members of the NuRD complex via IP-immunoblotting. Importantly, knockdown of two NuRD components restored somatic cell reprogramming and facilitated the exit from pluripotency, processes otherwise blocked by mTOR hyperactivation and TFE3 nuclear localisation.

The novel aspect of this study lies in uncovering the NuRD complex's role in mediating TFE3 functions in ESCs and somatic cell reprogramming. However, the exact mechanisms are not fully explored and current data primarily rely on quantifying reprogramming efficiency and exit from pluripotency.

1. NuRD depletion promotes differentiation under mTOR hyperactivation, contradicting earlier findings that NuRD complex is essential for ESC differentiation and KO of NuRD (e.g. Mbd3-KO) prevents differentiate. The authors should replicate NuRD KD in WT OG2-GFP ESCs for comparison with Tsc1-KO/TFE3-ERT models.

2) The authors assess pluripotency exit using a 96h differentiation protocol (2iL withdrawal), followed by 72h re-culture in 2iL medium. This approach complicates interpretation, as it combines exit from pluripotency (96h withdrawal) with reprogramming to a naïve state (72h re-culture). To clarify, the authors should present data for the 96h withdrawal as well. Additionally, qRT-PCR for naïve and primed markers is recommended to better characterize how NuRD complex KD rescues differentiation in Tsc1-KO ESCs.

3) The convergence of TFE3 and the NuRD complex on target genes remains unexplored. The authors should integrate their TFE3 ChIP-seq data with published NuRD ChIP-seq datasets from somatic cell reprogramming to strengthen their findings.

Additionally, performing Cut&Run assays to analyse TFE3 chromatin binding in Tsc1 KO ESCs and comparing these sites with available MBD3 ChIP-seq data would be important to provide further insight into their potential co-regulation of target genes.

4) The authors used IP-immunoblotting to confirm TFE3-NuRD interactions in TFE3-ERT cells, where TFE3 exhibits increased nuclear localization. Are these interactions dependent on mTOR activation? To clarify this, the authors should validate some of the TFE3 IP-immunoblotting in Tsc1-KO cells treated with Rapamycin, similar to the experiment shown in Fig EV2-A. This would help determine whether mTOR inhibition disrupts TFE3-NuRD interactions.

Referee #2:

In this study, Li et al. investigate the role of mTORC1 signaling in regulating stem cell fate transitions, focusing on its downstream effector, TFE3. They demonstrate that TFE3 inhibits cell reprogramming by recruiting the NuRD corepressor complex to regulate genes critical for cell fate transitions. The data presented in the manuscript appear robust and effectively support the authors' proposed model. While the study provides significant insights into the functions of TFE3, our comments focus on addressing remaining weaknesses and clarifying specific aspects to further strengthen the manuscript.

Major comments:

1. mTORC1 is a pleiotropic factor with many diverse effects. The authors discuss its nuanced role in various contexts related to pluripotency and reprogramming, but the manuscript lacks a more nuanced explanation in regard to autophagy. Specifically, mTOR is a critical negative regulator of general autophagy, while it may also regulate positively certain autophagic processes, such as the clearance of damaged mitochondria. Further clarification is recommended (in the introduction or discussion) to explain the present phenotype in a broader context of canonical mTORC1 function. (see also comment 4)
2. The sequencing data presented provide a strong foundation for the model. The authors could strengthen their conclusions by leveraging existing public datasets for the NuRD complex, beyond HDAC2, focusing on ESCs or MEFs (e.g. on GEO: MBD2/3 ChIP from GSE79770). Showing the target genes being bound by several members of the NuRD complex would be more convincing.
3. The manuscript could benefit from further analyses of the downstream targets at the individual gene level. For example, they should show more clearly the extent to which the downstream genes are similarly (or not?) regulated by Tsc1 KD vs Tfe3 shRNA vs NaB or KD of the NuRD complex partners (not just HDAC2 KO). This can be done by RT-qPCR analysis of target genes in the various models used. Additionally, related to comment #2, exploring public datasets to support the conclusion that these genes are targeted by the NuRD complex would further strengthen the findings.
4. Related to comment #3, the claim that mTORC1-TFE3-NuRD axis regulates autophagy/lysosome functions seem mostly relying on previous literature and GO analyses. Given that this point is mentioned throughout the manuscript, and even in their model figure, it should be better substantiated by experimental data. The authors should at least demonstrate that autophagy/lysosome are in fact affected in their various models (Tsc1 KO or Tfe shRNA for instance). This can be done by western blot or immunofluorescence of relevant markers for instance.
5. Why were IPs showing the interaction with the NuRD complex (Fig3B and 4A) done only using the TFE3-overexpressing cells? Overexpression can lead to artefacts. Could the IPs be repeated with WT cells, possibly using shTFE3 as a negative control?

Minor comments:

1. The labels of the KO/shRNAs are sometimes confusing. For instance, on Fig 2D, the label could be mistaken as "TFE3 KO" (instead of TFE3 signal in Tsc1 KO). We advise to clarify (here and in all similar panels) the model vs the technique used to avoid confusion from the readers.
2. Fig 1B: wouldn't mTOR activity be shown more directly by phospho-mTOR? Alternatively, showing phosphorylation of another downstream target of mTORC1 would be more convincing than only showing S6.
3. In RNA-seq analyses (Fig 1D/E & line 104) and other sequencing approaches, it is unclear if the p-value is corrected for multi-testing. This should be done (if not already done). If the p-value is already adjusted, it should be clearly stated.
4. Line 106: the authors state that there is "significant" overlap but no statistics are provided. If the authors meant the word in a colloquial way, we encourage to replace "significant" by another word to avoid confusion.
5. The pictures of the IFs are very small (see Fig 1H). It is very hard to see by eye if it is truly a change in location for TFE3 or a broader change in morphology of the cells. Zooming in more or showing bigger panels would be helpful.

6. The sequencing tracks are too small to be viewed properly (Fig. 2G, 4J). The window height should be increased.

7. Line 225-228: the end of the sentence appears to be missing. ("Given the context-dependent roles of the NuRD complex in reprogramming and the well-established reprogramming-enhancing effects of HDAC inhibition" ...?)

Referee #3:

In this work, Li, Xu, and colleagues show that Tsc1 loss and TFE3 nuclear translocation lead to stabilization of cell states, thereby impeding differentiation as well as reprogramming. They show that TFE3 cooperates with the NuRD complex in this role.

The paper is related to several previously published papers showing the defining role of TFE3 in maintaining pluripotency. It also brings a fresh perspective by assigning it a role in the transition into pluripotency during reprogramming.

Overall the paper is quite interesting, the figures are clear, logic is presented well. There are however deficiencies in data quality, availability, and interpretation that preclude immediate publication. I detail these points below. I would support publication if the authors are able to address major concerns.

Major:

- It is clear that Tsc1 KO leads to defects in OSKM reprogramming and shTFE3 rescues this. It is however hard to connect the dots here. The authors assume that the link in between is mTOR. mTOR activity is increased as expected upon Tsc1 loss here. Literature shows that TFE3 subcellular localization is affected by its phosphorylation by mTOR. Common knowledge goes that when mTOR is active, it phosphorylates TFE3, which keeps it cytoplasmic. When mTOR activity is reduced, TFE3 is hypophosphorylated and translocates to the nucleus (shown in several papers including Mathieu et al cited by the authors). Alternatively, AMPK has been shown to phosphorylate TFE3 when mTOR activity is inhibited. Either way, low activity of mTOR is associated with TFE3 nuclear localization. Some studies show that mTOR hyperactivity leads to nuclear TFE3, but careful analysis showed that in this case lysosome-localized mTOR is decreased, which still leads to reduced TFE3 phosphorylation, and thus its nuclear localization (Zwakenberg et al, 2024). The authors should check TFE3 phosphorylation under their assay conditions and test its dependency on mTOR activity (and not just Tsc1 absence) if they are to claim a link between mTOR and TFE3 in this context.

- TFE3 plays a critical role in the cytoplasm during pluripotency exit (shown e.g. in Villegas et al). While the authors show that there is nuclear binding of TFE3 and that shTFE3 rescues pluripotency transition defects, there is no direct proof as to the nonessentiality of the cytoplasmic function in the mentioned transitions. In Fig 1h and EV1A the authors score most TFE3 signal in shTsc2 as nuclear, however it looks more like nuclear+cytoplasmic to me. To make a strong point about the essentiality of the nuclear function, the authors should selectively inhibit nuclear or cytoplasmic functions (e.g. via the point mutant described in Villegas et al).

- The authors map TFE3 binding in OSKM. A huge number of peaks is detected, spanning nearly all genes (>19000). Likewise for HDAC2 binding in Fig4G. Since the sites were mapped by CUT&Tag, which is prone to spurious signal at promoters, the authors need to crosscheck that this is real occupancy signal. Two ways I suggest are C&T under TFE3 KO/KD or ChIP-seq in comparison with C&T.

- I could not find C&T datasets in this paper. The data are not deposited to GEO or a similar server. The datasets EV1/2 were not reachable. Thus it is hard to interpret the reliability of these data.

- Additionally it is notable that the more robustly bound sites are not affected in the KO, both for TFE3 and HDAC2. What is the authors' explanation for the large number of sites remaining unchanged, despite the large nuclear/cytoplasmic shifts in the KO?

Minor:

- Please show individual data points in box plots and where relevant.

- Are the IFs performed once? The graphs look like single cell quantifications from one biological replicate are plotted. If so, a second replicate needs to be performed. Also please indicate the precise number of data points, and not just $n > 100$ everywhere.

- I suggest to move Fig 4 to Fig 3 (and vice versa) so that the flow is more coherent.

Guangzhou Institutes of Biomedicine and Health (GIBH)
Chinese Academy of Sciences (CAS)

May 4th, 2025

Dear Achim,

I would like to thank you and the reviewers for the positive assessment of our manuscript. We also feel grateful for providing us with an extension to finish our revision. We have now revised the manuscript in response to the comments of the reviewers and added new experimental and sequencing data that strongly support our original claims.

Please find a detailed point-by-point response to the reviewers' comments below.

Sincerely,

Xueting Xu

Xueting Xu on behalf of all authors

Guangzhou Institutes of Biomedicine and Health, Chinese Academy of Sciences, Guangzhou 510530, China.

e-mail: xu_xueting@gibh.ac.cn

Referee #1:

Li et al demonstrate that hyperactivation of mTOR signalling and nuclear localisation of TFE3 promote self-renewal and inhibit the exit from pluripotency, consistent with prior findings. Additionally, hyperactivation of mTOR signalling and nuclear localisation of TFE3 prevent somatic cell reprogramming. To explore TFE3's nuclear functions, the authors analysed published IP-MS data for TFE3 interactors and confirmed its direct interaction with members of the NuRD complex via IP-immunoblotting. Importantly, knockdown of two NuRD components restored somatic cell reprogramming and facilitated the exit from pluripotency, processes otherwise blocked by mTOR hyperactivation and TFE3 nuclear localisation.

The novel aspect of this study lies in uncovering the NuRD complex's role in mediating TFE3 functions in ESCs and somatic cell reprogramming. However, the exact mechanisms are not fully explored and current data primarily rely on quantifying reprogramming efficiency and exit from pluripotency.

Response: We thank the reviewer for the positive assessment of our manuscript. For the mechanism, we have performed new sequencing and analysis that support better of our previous conclusion. Please see details below.

1) NuRD depletion promotes differentiation under mTOR hyperactivation, contradicting earlier findings that NuRD complex is essential for ESC differentiation and KO of NuRD (e.g. Mbd3-KO) prevents differentiate. The authors should replicate NuRD KD in WT OG2-GFP ESCs for comparison with Tsc1-KO/TFE3-ERT models.

Response: The reviewer raised relevant point. We went through earlier literatures, and found that the conclusions are inconsistent. Some studies reported that lacking NuRD components Mbd3 failed to commit to developmental lineages in ESCs (Kaji K et al, 2006; Reynolds N et al, 2012), or mESCs lacking all three MTA proteins (MTA1/2/3) failed to pluripotency exit (Burgold T et al, 2019); others, however, reported that *Gatad2b* knockout ESCs could maintain normal differentiation potential (Wang B et al, 2021), and silencing of *Chd4* upregulated differentiation-associated genes during ESCs differentiation (Zhao H et al, 2017). These studies suggest a context- or

experimental condition- dependent roles of NuRD in differentiation. We knocked down *Gatad2a* and *Mbd3* in WT-OG2 mESCs, and didn't find significant influence on the exit of pluripotency (Fig EV2G, Fig EV3C), which could confirm our finding that NuRD depletion promotes differentiation under mTORC1 hyperactivation or nuclear induction of ectopic TFE3.

2) The authors assess pluripotency exit using a 96h differentiation protocol (2iL withdrawal), followed by 72h re-culture in 2iL medium. This approach complicates interpretation, as it combines exit from pluripotency (96h withdrawal) with reprogramming to a naïve state (72h re-culture). To clarify, the authors should present data for the 96h withdrawal as well. Additionally, qRT-PCR for naïve and primed markers is recommended to better characterize how NuRD complex KD rescues differentiation in Tsc1-KO ESCs.

Response: We agreed with the reviewer that the results from the 96h-72h system could be complicated by distinct mechanisms. In the new revision, we first re-analyzed the published RNA-seq data of TFE3-ERT 34h-exit (Villegas F et al, 2019), and found that, after 34h 2iL withdraw, while in the WT cells the naïve pluripotency genes were downregulated with the primed pluripotency genes upregulated, TFE3 OE (TFE3-ERT+Tamoxifen) failed to switch in the two groups of genes, indicating a blockade of transcriptional changes during exit of (naïve) pluripotency, though these TFE3 OE cells in 2iL already fell from an optimal naïve pluripotent state (Fig EV4A).

In our models, we now provided both imaging data (Fig EV3) and RT-qPCR results of naïve and primed pluripotency genes (Fig EV4B, C) for the cells of 96h withdrawal of 2iL (exit of pluripotency). On the one hand, as shown in the images, WT cells exhibited extensive cell death following pluripotency exit, with almost no GFP⁺ cells remaining, but KO and TFE3-ERT overexpression cells retained a significant number of GFP⁺ cells, moreover, knockdown of NuRD complex components *Gatad2a* and *Mbd3* reduced the number of GFP⁺ cells in both KO and TFE3-ERT overexpressing cells, indicating a rescue of the blocked pluripotency exit phenotype (Fig EV3D, E). On the other hand, RT-qPCR analysis revealed that although the expression of naïve pluripotency genes was higher in KO cells compared to WT cells after 96 hours exit, both showed a marked decrease relative to 2iL-maintained WT mESCs, and knockdown of NuRD complex components did not rescue the expression of naïve genes (Fig EV4B). In contrast, primed pluripotency gene expression was

significantly upregulated after 96 hours exit in WT cells but not in KO cells, and this repression could be alleviated by knockdown of NuRD components (Fig EV4C). These results suggest a switch in gene expression from naïve to primed pluripotency markers during the exit from pluripotency. Importantly, the NuRD complex appears to play a key role in repressing primed pluripotency gene expression, but not naïve gene expression. This supports our conclusion that the TFE3-NuRD interaction directly inhibits the activation of primed pluripotency genes during the exit from pluripotency.

3) *The convergence of TFE3 and the NuRD complex on target genes remains unexplored. The authors should integrate their TFE3 ChIP-seq data with published NuRD ChIP-seq datasets from somatic cell reprogramming to strengthen their findings. Additionally, performing Cut&Run assays to analyse TFE3 chromatin binding in Tsc1 KO ESCs and comparing these sites with available MBD3 ChIP-seq data would be important to provide further insight into their potential co-regulation of target genes.*

Response: Thanks for the comments. In this revision, we re-did RNA-seq of *Tfe3* knockdown rescued cells during reprogramming (Fig 3F, G), and consider the variation of different context, also to exclude the effect of cytoplasmic protein, we re-did TFE3 and RBBP4 CUT&Tag after nuclear-cytoplasmic fractionation in *Tsc1* WT, KO, KO+rapamycin (for further confirm target genes regulated by mTORC1 activation) reprogramming cells (day 10) (Fig 4A-D; Fig EV8) and *Tsc1* WT, KO mESCs (both in 2iL mESCs and N2B27 exit 40h cells) (Fig EV5, Fig EV6). On the one hand, we found that both TFE3 and NuRD bind and repress pluripotency genes (*Esrrb*, *Nr5a2*, and *Dnmt3a* etc...) during reprogramming (Fig 4D). Together with the results in the exit of pluripotency context (Fig EV5, Fig EV6), our findings indicated that TFE3-NuRD association represses the induction of genes critical for pluripotent cell fate transitions.

4) *The authors used IP-immunoblotting to confirm TFE3-NuRD interactions in TFE3-ERT cells, where TFE3 exhibits increased nuclear localization. Are these interactions dependent on mTOR activation? To clarify this, the authors should validate some of the TFE3 IP-immunoblotting in Tsc1-KO cells treated with Rapamycin, similar to the experiment shown in Fig EV2-A. This would help determine whether mTOR inhibition disrupts TFE3-NuRD interactions.*

Response: The reviewer raised a good point. We repeated co-IP experiments

after nuclear-cytoplasmic fractionation in *Tsc1* WT, KO, and KO+rapamycin cells both in 2iL mESCs (Fig 2G) and reprogramming day 5 (Fig 3B), these results show that the TFE3-NuRD association increased in *Tsc1* KO vs WT cells, and rescued by rapamycin. Furthermore, our new CUT&Tag sequencing results also confirmed that rapamycin could significantly rescue the co-binding and gene repression from TFE3 and NuRD in *Tsc1* KO reprogramming (Fig. 4A-D). These new results confirm that *Tsc1* KO induced mTORC1 hyperactivation underlies nuclear TFE3-NuRD association and repression of cell fate transitions.

Referee #2:

In this study, Li et al. investigate the role of mTORC1 signaling in regulating stem cell fate transitions, focusing on its downstream effector, TFE3. They demonstrate that TFE3 inhibits cell reprogramming by recruiting the NuRD corepressor complex to regulate genes critical for cell fate transitions. The data presented in the manuscript appear robust and effectively support the authors' proposed model. While the study provides significant insights into the functions of TFE3, our comments focus on addressing remaining weaknesses and clarifying specific aspects to further strengthen the manuscript.

Response: We thank the reviewer for positive comments. We have now provided additional experimentations, analysis, and clarifications that support better our hypotheses.

Major comments:

1. mTORC1 is a pleiotropic factor with many diverse effects. The authors discuss its nuanced role in various contexts related to pluripotency and reprogramming, but the manuscript lacks a more nuanced explanation in regard to autophagy. Specifically, mTOR is a critical negative regulator of general autophagy, while it may also regulate positively certain autophagic processes, such as the clearance of damaged mitochondria. Further clarification is recommended (in the introduction or discussion) to explain the present phenotype in a broader context of canonical mTORC1 function. (see also comment 4).

Response to this comment #1 and comment #4 below from the same reviewer: We thank the reviewer for raising this important point. In this study, we observed that hyperactivation of mTORC1 induces ectopic nuclear localization of TFE3 and upregulates the expression of lysosome- and autophagy-related genes. However, despite this transcriptional activation, autophagy is ultimately inhibited during both reprogramming (Fig 5e in Wu Y et al, 2015) and pluripotency exit (see figure below). Due to time constraints, we did not directly detect mitophagy, and therefore this remains a hypothesis. Consequently, we have decided to remove schematic B and retain this point only in the discussion section. Moreover, previous studies have shown that mTORC1 hyperactivation during reprogramming leads to increased

mitochondrial mass and elevated ROS levels (Fig 5g-i in Wu Y et al, 2015). We speculate that even if mitophagy activated, it may be insufficient to counteract the oxidative stress caused by damaged mitochondria. These results giving the possibility that there has another undiscovered barrier that disconnects lysosome/autophagy gene expression from effective autophagic activation under conditions of mTORC1 hyperactivation. We have clarified this point in the revised discussion and hope this addresses the reviewer's concern.

Figure. mTORC1 hyperactivation inhibited autophagy during pluripotency exit.

2. The sequencing data presented provide a strong foundation for the model. The authors could strengthen their conclusions by leveraging existing public datasets for the NuRD complex, beyond HDAC2, focusing on ESCs or MEFs (e.g. on GEO: MBD2/3 ChIP from GSE79770). Showing the target genes being bound by several members of the NuRD complex would be more convincing.

Response: Thanks for the comments. In this revision, considering the common inconsistency among different reprogramming conditions, we performed RNA-seq in *Tfe3*-knockdown rescued cells during reprogramming (Fig 3F, G), and TFE3 and RBBP4 (a specific component of NuRD complex) CUT&Tag. The CUT&Tag was done in *Tsc1* WT, KO, and KO+Rapamycin cells after nuclear-cytoplasmic fractionation for reprogramming at day 10 (Fig 4A-D, Fig EV8) and mESCs in both 2iL and exit (N2B27, 40h) (Fig EV5, Fig EV6). Consistent with our last vision, we found that the TFE3 and NuRD co-bind and repress pluripotency genes (*Esrrb*, *Nr5a2*, and *Dnmt3a* etc...) during reprogramming (Fig 4D), and

inhibited stem cell differentiation/development related genes during pluripotency exit (Fig EV5B, D). Furthermore, NuRD complex components *Gatad2a/Mbd3* knockdown could rescue the expression level of *Esrrb* in reprogramming (Fig 3I) and primed pluripotency genes in pluripotency exit (Fig EV4C). These results strengthen our findings that TFE3-NuRD association transcriptional repress genes expression critical for pluripotent cell fate transitions.

3. *The manuscript could benefit from further analyses of the downstream targets at the individual gene level. For example, they should show more clearly the extent to which the downstream genes are similarly (or not?) regulated by Tsc1 KD vs Tfe3 shRNA vs NaB or KD of the NuRD complex partners (not just HDAC2 KO). This can be done by RT-qPCR analysis of target genes in the various models used. Additionally, related to comment #2, exploring public datasets to support the conclusion that these genes are targeted by the NuRD complex would further strengthen the findings.*

Response: Thanks for the comments. We think the above new sequencing data and analysis results should be sufficient also to answer this question, and please see our response to the above comment #2.

4. *Related to comment #3, the claim that mTORC1-TFE3-NuRD axis regulates autophagy/lysosome functions seem mostly relying on previous literature and GO analyses. Given that this point is mentioned throughout the manuscript, and even in their model figure, it should be better substantiated by experimental data. The authors should at least demonstrate that autophagy/lysosome are in fact affected in their various models (Tsc1 KO or Tfe shRNA for instance). This can be done by western blot or immunofluorescence of relevant markers for instance.*

Response: Please see our response to the above comment #1.

5. *Why were IPs showing the interaction with the NuRD complex (Fig3B and 4A) done only using the TFE3-overexpressing cells? Overexpression can lead to artefacts. Could the IPs be repeated with WT cells, possibly using shTFE3 as a negative control?*

Response: We agree with the reviewer. We repeated co-IP immunoblotting after nuclear-cytoplasmic fractionation in *Tsc1* WT, KO, KO+rapamycin cells both in 2iL mESCs (Fig 2G) and reprogramming day 5 (Fig 3B), these results show that TFE3-NuRD interactions were increased in *Tsc1* KO cells and could be rescued by rapamycin, confirm that TFE3-NuRD interactions but the association increased dependent on mTORC1 hyperactivation.

Minor comments:

1. *The labels of the KO/shRNAs are sometimes confusing. For instance, on Fig 2D, the label could be mistaken as "TFE3 KO" (instead of TFE3 signal in Tsc1 KO). We advise to clarify (here and in all similar panels) the model vs the technique used to avoid confusion from the readers.*

Response: We are sorry for the confusion. In the current revision, we have modified and clarified our labels, please see new Fig 4D.

2. *Fig 1B: wouldn't mTOR activity be shown more directly by phospho-mTOR? Alternatively, showing phosphorylation of another downstream target of mTORC1 would be more convincing than only showing S6.*

Response: Thanks for the comments. In the revision, we added phosphorylation of S6K and 4EBP1, other more direct downstream targets of mTORC1, as indicators of mTORC1 activation (Fig 1B, Fig EV1B, Fig EV2A).

3. *In RNA-seq analyses (Fig 1D/E & line 104) and other sequencing approaches, it is unclear if the p-value is corrected for multi-testing. This should be done (if not already done). If the p-value is already adjusted, it should be clearly stated.*

Response: Sorry for our unclear description, *P*-value was adjusted, and we stated it clearly in this current revision.

4. *Line 106: the authors state that there is "significant" overlap but no statistics are provided. If the authors meant the word in a colloquial way, we encourage*

to replace "significant" by another word to avoid confusion.

Response: Thanks for the comments. We have replaced "significant" to "remarkable" (Line 104).

5. The pictures of the IFs are vey small (see Fig 1H). It is very hard to see by eye if it is truly a change in location for TFE3 or a broader change in morphology of the cells. Zooming in more or showing bigger panels would be helpful.

Response: We thank the reviewer for raising this point. Zooming in more of this IFs pictures were did in this current revision.

6. The sequencing tracks are too small to be viewed properly (Fig. 2G, 4J). The window height should be increased.

Response: We thank the reviewer for mention this point. The increased window height is shown in Fig 4D.

7. Line 225-228: the end of the sentence appears to be missing. ("Given the context-dependent roles of the NuRD complex in reprogramming and the well-established reprogramming-enhancing effects of HDAC inhibition" ...?).

Response: Sorry for mistake. We have corrected it in the revised text (Line 259-264).

Referee #3:

In this work, Li, Xu, and colleagues show that Tsc1 loss and TFE3 nuclear translocation lead to stabilization of cell states, thereby impeding differentiation as well as reprogramming. They show that TFE3 cooperates with the NuRD complex in this role.

The paper is related to several previously published papers showing the defining role of TFE3 in maintaining pluripotency. It also brings a fresh perspective by assigning it a role in the transition into pluripotency during reprogramming.

Overall the paper is quite interesting, the figures are clear, logic is presented well. There are however deficiencies in data quality, availability, and interpretation that preclude immediate publication. I detail these points below. I would support publication if the authors are able to address major concerns.

Response: We thank the reviewer for the positive assessment of our manuscript. We have now provided additional experimentations, analysis, and clarifications that support better our hypotheses.

Major:

- It is clear that Tsc1 KO leads to defects in OSKM reprogramming and shTFE3 rescues this. It is however hard to connect the dots here. The authors assume that the link in between is mTOR. mTOR activity is increased as expected upon Tsc1 loss here.

Literature shows that TFE3 subcellular localization is affected by its phosphorylation by mTOR. Common knowledge goes that when mTOR is active, it phosphorylates TFE3, which keeps it cytoplasmic. When mTOR activity is reduced, TFE3 is hypophosphorylated and translocates to the nucleus (shown in several papers including Mathieu et al cited by the authors). Alternatively, AMPK has been shown to phosphorylate TFE3 when mTOR activity is inhibited. Either way, low activity of mTOR is associated with TFE3

nuclear localization. Some studies show that mTOR hyperactivity leads to nuclear TFE3, but careful analysis showed that in this case lysosome-localized mTOR is decreased, which still leads to reduced TFE3 phosphorylation, and thus its nuclear localization (Zwakenberg et al, 2024). The authors should check TFE3 phosphorylation under their assay conditions and test its dependency on mTOR activity (and not just Tsc1 absence) if they are to claim a link between mTOR and TFE3 in this context.

Response: We thank the reviewer for raising this point. As the study by Zwakenberg et al did not show specific phosphorylation sites of TFE3, we examined our own unpublished phosphoproteomic sequencing data on TFE3 phosphorylation in 2iL-cultured mESCs (from another unpublished manuscript, see below). Consistent with the conclusions of Zwakenberg et al, we observed that several phosphorylation sites on TFE3 were downregulated upon mTORC1 hyperactivation, and this effect could be reversed by rapamycin treatment.

Protein_names	Gene_names	Tsc1 KO/WT	P value	KO+Rapamycin/KO	P value
Transcription factor E3	Tfe3_1_S_510	-0.623435488	0.13862408	9.614754254	9.81E-07
Transcription factor E3	Tfe3_2_S_516	-0.623435488	0.13862408	8.312611	1.76E-05
Transcription factor E3	Tfe3_3_S_518	-7.57997177	5.74E-07	9.122951032	2.98E-07
Transcription factor E3	Tfe3_4_S_522	-7.57997177	5.74E-07	7.955010382	1.60E-06
Transcription factor E3	Tfe3_5_S_529	-0.623435488	0.13862408	1.363737633	0.146495237
Transcription factor E3	Tfe3_6_S_530	-8.114635896	1.93E-07	7.5101972	5.63E-06

- TFE3 plays a critical role in the cytoplasm during pluripotency exit (shown e.g. in Villegas et al). While the authors show that there is nuclear binding of TFE3 and that shTFE3 rescues pluripotency transition defects, there is no direct proof as to the nonessentiality of the cytoplasmic function in the mentioned transitions. In Fig 1h and EV1A the authors score most TFE3 signal in shTsc2 as nuclear, however it looks more like nuclear+cytoplasmic to me. To make a strong point about the essentiality of the nuclear function, the authors should selectively inhibit nuclear or cytoplasmic functions (e.g. via the point mutant described in Villegas et al).

Response: In the study by Villegas et al (2019, Fig 7B, C), the authors showed that a TFE3 cytoplasmic sequestration mutant (exon 3-end) did not affect pluripotency exit. In our current revision, we overexpression TFE3-ERT in both reprogramming cells (Fig EV1D, E) and mESCs (Fig EV2E), Without tamoxifen treatment, TFE3 remained sequestered in the cytoplasm (Fig EV1E, Fig EV2E) and did not influence reprogramming efficiency or pluripotency exit

(Fig EV1F, Fig EV2F). These results support our conclusion that only nuclear-localized TFE3 plays a functional role in pluripotent cell fate transitions. Regarding the immunofluorescence quantification, the reviewer is correct. In the TFE3 IF images, most signals in *Tsc1* knockout or *Tsc2* knockdown cells were nuclear+cytoplasmic, we re-count these signals and modify it in this current revision (Fig 1H, Fig EV1A, E and Fig EV2B, E).

- *The authors map TFE3 binding in OSKM. A huge number of peaks is detected, spanning nearly all genes (>19000). Likewise for HDAC2 binding in Fig4G. Since the sites were mapped by CUT&Tag, which is prone to spurious signal at promoters, the authors need to crosscheck that this is real occupancy signal. Two ways I suggest are C&T under TFE3 KO/KD or ChIP-seq in comparison with C&T.*

Response: We agree with this reviewer. In this revision, to exclude the effect of cytoplasmic protein, we re-did TFE3 and RBBP4 (another NuRD complex component) cut-tag after nuclear-cytoplasmic fractionation in *Tsc1* WT, KO, KO+rapamycin reprogramming cells (day 10) (Fig 4A-D, Fig EV8) and *Tsc1* WT, KO mESCs (both in 2iL mESCs and N2B27 exit 40h cells) (Fig EV5, Fig EV6). Consistent with our last vision, we found that the TFE3-NuRD nuclear association binds and represses pluripotency genes (*Esrrb*, *Nr5a2*, and *Dnmt3a* etc...) during reprogramming (Fig 4D), and inhibits stem cell differentiation/development related genes during pluripotency exit (Fig EV5B, D). Furthermore, NuRD complex components *Gatad2a/Mbd3* knockdown could rescue the expression level of *Esrrb* in reprogramming (Fig 3I). These results strengthen our findings that TFE3-NuRD association transcriptionally represses genes expression critical for pluripotent cell fate transitions.

- *I could not find C&T datasets in this paper. The data are not deposited to GEO or a similar server. The datasets EV1/2 were not reachable. Thus it is hard to interpret the reliability of these data.*

Response: Sorry for this missing datasets GEO number, we already add it in the current revision.

- *Additionally it is notable that the more robustly bound sites are not affected in the KO, both for TFE3 and HDAC2. What is the authors' explanation for the*

large number of sites remaining unchanged, despite the large nuclear/cytoplasmic shifts in the KO?

Response: We think although TFE3 and NuRD binding with one genes on several peaks, but only little peaks were important for gene expression, this is why a large number of sites remaining unchanged.

Minor:

- Please show individual data points in box plots and where relevant.

Response: Individual data points in box plots and relevant were added in this current revision.

- Are the IFs performed once? The graphs look like single cell quantifications from one biological replicate are plotted. If so, a second replicate needs to be performed. Also please indicate the precise number of data points, and not just $n > 100$ everywhere.

Response: We did IFs twice expect Fig EV1E, new statistical chart and precise number were performed in this current revision.

- I suggest to move Fig 4 to Fig 3 (and vice versa) so that the flow is more coherent.

Response: We thank the reviewer for raising this point. Since we first found hyperactivation of mTORC1 blocks reprogramming through TFE3 nuclear translocation, then we found TFE3 associated with NuRD complex in mESCs, finally confirmed this association in reprogramming. Now we rearrangement our Figure with new data, hope the reviewer could be satisfied.

Dear Dr. Xu

Thank you for the submission of your revised manuscript to our editorial offices. I have now received the reports from the three referees that were asked to re-evaluate the study, you will find below. As you will see, the referees now support the publication of your manuscript in EMBO reports.

However, there are remaining issues. Referee #1 requests textual changes. More importantly, referee #3 remains unconvinced of some datasets. For the first point I would suggest to add this unpublished data. For the second point, a re-analysis, including negative controls, seems appropriate. Please do this in a final revised manuscript and also provide a final p-b-p-response addressing these concerns.

- Please have your final manuscript text proofread by a native speaker (see also the comments of referee #2).
- Please reduce the abstract to not more than 175 words.
- We now use CRediT to specify the contributions of each author in the journal submission system. CRediT replaces the author contribution section. Please use the free text box to provide more detailed descriptions and do NOT provide your final manuscript text file with an author contributions section. See also our guide to authors:
<https://www.embopress.org/page/journal/14693178/authorguide#authorshipguidelines>
- Please order the sections like this, using these names:
Title page - Abstract - Keywords - Introduction - Results - Discussion - Methods - Data availability section - Acknowledgements - Disclosure and Competing Interests Statement - References - Figure legends - Expanded View Figure legends
- Please provide individual production quality figure files as .eps, .tif, .jpg (one file per figure), of the main and EV figures. Please upload these as separate, individual files upon re-submission. Please re-arrange or combine the figures to have 5-6 final main and 5-6 final EV figures. Please also update any callouts after these changes.
- Please make sure that all figure panels (main and EV figures) are called out separately and sequentially. Presently, there seem to be no callouts for panels B and E of Fig. 4 (but a callout for 4G, which seems not to exist). Please check.
- Please check again that the number "n" for how many independent experiments were performed, their nature (biological versus technical replicates), the bars and error bars (e.g. SEM, SD) and the test used to calculate p-values is indicated in the respective figure legends. Please also check that all the p-values are explained in the legend, and that these fit to those shown in the figure. Please provide statistical testing where applicable. Please avoid the phrase 'independent experiment' but clearly state if these were biological or technical replicates. Please also indicate (e.g. with n.s.) if testing was performed, but the differences are not significant. In case n=2, please show the data as separate datapoints without error bars and statistics. See also:
<http://www.embopress.org/page/journal/14693178/authorguide#statisticalanalysis>
- If n<5, please show single datapoints for diagrams. Moreover:
 - Please note that the legend for subfigures of EV-5 is mislabeled in the manuscript. This needs to be rectified.
 - Please note that the legend for figure 3I is missing in the manuscript. This needs to be rectified.
 - Please note that the exact p values are not provided in the legends of figures 1C, EV1 C.
 - Please indicate what */ **/ ***/ **** represents; if this represents p value(s), please indicate the statistical test used and where appropriate, specify the exact p value in the legend(s) of figure(s) 1K, 2H-J; 3C, D, E, H, I; EV1 F, EV2 D, F, G-I; EV4 B, C; EV7A, B.
 - Please indicate the statistical test used for data analysis in the legends of figures 1F, 3G, EV5 B, D; EV6 A, B; EV8A.
 - Please note that information related to n is missing in the legend of figure 3I
 - Please note that the error bars are not defined in the legends of figures 1H, K; 3I, EV1 A, E; EV2 B, E
- Please add to each legend (main and EV figures, where applicable) a 'Data Information' section (or name the provided section like this) explaining the statistics used or providing information regarding replicates and scales. See:

- The Data Availability Section (DAS) is restricted to information regarding large primary datasets deposited at external databases. Thus, please remove all other text not related to deposited datasets from the DAS.
- All Materials and Methods need to be described in the main text using our 'Structured Methods' format, which is required for all

research articles. According to this format, the Methods section should include a Reagents and Tools Table (listing key reagents, experimental models, software, and relevant equipment and including their sources and relevant identifiers), uploaded as separate file, and a Methods section in which we encourage the authors to describe their methods using a step-by-step protocol format with bullet points, to facilitate the adoption of the methodologies across labs. More information on how to adhere to this format as well as downloadable templates (.doc) for the Reagents and Tools Table can be found in our author guidelines (section 'Structured Methods'):

- Please move the primer information (Tables 1 and 2, or EV1 and EV2) to the Reagents & Tools Table and remove the tables from the manuscript files. Please update any callouts.

In addition, I would need from you uploaded separately:

Best,

Referee #1:

The authors have addressed all of my comments. However, one point requires a minor textual revision: Most of the colony formation assays presented in the manuscript are based on measuring OCT4-positive surviving colonies. A reduction in colony number could result either from differentiation (i.e., loss of OCT4-GFP expression) or from cell death, both of which would lead to fewer OCT4-positive colonies (e.g., Fig. 2I, Fig. EV3). Therefore, the authors should moderate their conclusions regarding the differentiation-promoting effects of NuRD knockdown and clarify that the observed reductions could also reflect decreased cell viability.

Referee #2:

The authors have addressed my previous concerns to my satisfaction, and the overall quality of the work has improved. There are a few very minor typos or awkward phrasings, but nothing substantial. These can be easily fixed during the editorial process. I recommend the manuscript for publication.

Referee #3:

The authors added several new experiments in the revised manuscript. I appreciate the effort, however it seems the results are interpreted with the aim of getting to the desired results instead of seeing the data for what it is.

- The author's response to my first question includes unpublished data that is in no way traceable or verifiable by the reviewers.
- My original concerns also apply to the new CUT&Tag data in Figure 4. Too many peaks and gene targets are detected, there is no negative control, and most of the peaks that the authors highlight are very low enrichment. As it is, I cannot corroborate whether these are above the background.

As is, although there is a lot of data, some of the conclusions remain weak.

Guangzhou Institutes of Biomedicine and Health (GIBH)
Chinese Academy of Sciences (CAS)

July 6th, 2025

Dear Achim,

I would like to thank you and the reviewers for the positive evaluation of our manuscript. We are also grateful for the extension granted to complete our revision. We have now addressed the remaining concerns raised by the three reviewers (detailed below) and revised the manuscript in accordance with your suggestions and the journal's guidelines.

Please do not hesitate to contact us if any further changes are needed. We look forward to your response.

Sincerely,

Xueting Xu

Xueting Xu on behalf of all authors

Guangzhou Institutes of Biomedicine and Health, Chinese Academy of Sciences, Guangzhou 510530, China.

e-mail: xu_xueting@gibh.ac.cn

Referee #1:

The authors have addressed all of my comments. However, one point requires a minor textual revision:

Most of the colony formation assays presented in the manuscript are based on measuring OCT4-positive surviving colonies. A reduction in colony number could result either from differentiation (i.e., loss of OCT4-GFP expression) or from cell death, both of which would lead to fewer OCT4-positive colonies (e.g., Fig. 2I, Fig. EV3). Therefore, the authors should moderate their conclusions regarding the differentiation-promoting effects of NuRD knockdown and clarify that the observed reductions could also reflect decreased cell viability.

Response: We thank the reviewer for raising this important point. We have addressed it in the Discussion section (lines 380-388), and we hope this clarification meets the reviewer's expectations.

Referee #2:

The authors have addressed my previous concerns to my satisfaction, and the overall quality of the work has improved. There are a few very minor typos or awkward phrasings, but nothing substantial. These can be easily fixed during the editorial process. I recommend the manuscript for publication.

Response: We thank the reviewer for the positive assessment of our manuscript. We apologize for the previous errors and inappropriate phrasing, which have now been corrected in the current revision.

Referee #3:

The authors added several new experiments in the revised manuscript. I

appreciate the effort, however it seems the results are interpreted with the aim of getting to the desired results instead of seeing the data for what it is.

- The author's response to my first question includes unpublished data that is in no way traceable or verifiable by the reviewers.

- My original concerns also apply to the new CUT&Tag data in Figure 4. Too many peaks and gene targets are detected, there is no negative control, and most of the peaks that the authors highlight are very low enrichment. As it is, I cannot corroborate whether these are above the background.

As is, although there is a lot of data, some of the conclusions remain weak.

Response: We thank the reviewer for the positive assessment of our work and for raising these two important points.

Regarding the first point, since this information is not included in the manuscript and involves unpublished data, we are unable to provide the full sequencing dataset due to confidentiality concerns. However, to support the reliability of our findings, we are willing to share additional unpublished data. Specifically, we performed phospho-proteomic analyses under two distinct conditions: (1) treatment of *Tsc1* KO mESCs with a high dose of rapamycin (20 nM) for 2 hours to capture the immediate downstream targets of mTORC1, and (2) treatment with a low dose of rapamycin (0.3 nM) for 3 days to identify targets that can be eventually rescued. In both conditions, we observed decreased phosphorylation at S518, S522, and S530 of TFE3 in *Tsc1* KO cells, which was restored by rapamycin treatment. Because these phosphorylation sites are non-canonical and lack commercially available antibodies for validation by western blot, we instead assessed classical mTORC1 targets-phospho-RPS6 and phospho-EIF4EBP1-and confirmed that their phosphorylation levels increased upon mTORC1 hyperactivation and were rescued by rapamycin in both conditions. These results validate the reliability of our phospho-proteomics data and support the conclusion that mTORC1 hyperactivation leads to reduced

phosphorylation of TFE3.

Regarding the second point, we have now included appropriate negative controls to demonstrate that the detected peaks represent true signals rather than background noise (Fig 3A, C & Fig5A, D). Additionally, we performed a rescue experiment by overexpressing *Lin28a*, which effectively restored the blocked pluripotency exit caused by TFE3 nuclear translocation (Fig 3D-G), confirming the reliability of our data and analyses.

We hope this addresses the reviewer's concerns.

Gene_names	Tsc1 KO/WT	P values	Tsc1 KO+rapamycin 20nM (2 hours)/KO	P values	Tsc1 KO+rapamycin 0.3nM (3 days)/KO	P values
Tfe3 S_510	-0.623435488	0.138624	0.613064061	0.095434	9.614754254	9.81E-07
Tfe3 S_516	-0.623435488	0.138624	0.613064061	0.095434	8.312611	1.76E-05
Tfe3 S_518	-7.57997177	5.74E-07	8.101186624	9.55E-07	9.122951032	2.98E-07
Tfe3 S_522	-7.57997177	5.74E-07	8.101186624	9.55E-07	7.955010382	1.60E-06
Tfe3 S_529	-0.623435488	0.138624	6.159940831		1.363737633	0.146495
Tfe3 S_530	-8.114635896	1.93E-07	6.84881441	8.67E-05	7.5101972	5.63E-06
Rps6 S_235	3.206948309	0.035862	-13.5571877	0.015335	-12.80651413	0.01534
Eif4ebp1 T_37	9.671905271	8.83E-07	-9.682276699	0.002022	-8.931603127	0.002192
Eif4ebp1 T_46	12.3897222	0.042767	-12.40009363	0.002022	-11.64942005	0.001254

All editorial and formatting issues were resolved by the authors.

Dr. Xueting Xu

Guangzhou Institutes of Biomedicine and Health, Chinese Academy of Sciences
Guangdong Provincial Key Laboratory of Stem Cell and Regenerative Medicine
190 Kaiyuan Avenue, Guangzhou Science Park, Luogang District, Guangzhou
Guangdong 510530
China

Dear Dr. Xu,

Thank you for the submission of your further revised manuscript to our editorial offices. Going through your p-b-p-response and the revised manuscript, I consider the remaining points of the referees as adequately addressed.

I am thus very pleased to accept your manuscript for publication in the next available issue of EMBO reports. Thank you for your contribution to our journal.

Yours sincerely,
